# Combined Thermal and Colorimetric Analysis as a Tool for Detecting Counterfeit Viagra^®^ Tablets

**DOI:** 10.3390/ph19010078

**Published:** 2025-12-30

**Authors:** Paweł Ramos, Sławomir Wilczyński, Klaudia Stocerz, Roman Adamczyk, Anita Stanjek-Cichoracka

**Affiliations:** 1Department of Community Pharmacy, Faculty of Pharmaceutical Sciences in Sosnowiec, Medical University of Silesia, Jedności 8b, 41-200 Sosnowiec, Poland; klaudia.stocerz@sum.edu.pl (K.S.); roman.adamczyk@sum.edu.pl (R.A.); anita.stanjek@sum.edu.pl (A.S.-C.); 2Department of Basic Biomedical Sciences, Faculty of Pharmaceutical Sciences in Sosnowiec, Medical University of Silesia, Jedności 8b, 41-200 Sosnowiec, Poland; swilczynski@sum.edu.pl; 3Doctoral School, Medical University of Silesia, Poniatowskiego 15, 40-055 Katowice, Poland

**Keywords:** sildenafil, Viagra^®^, thermal analysis, colorimetry, CIE L*a*b*, counterfeit drugs, darknet

## Abstract

**Background/Objectives**: This study aimed to perform a comparative analysis of the original Viagra^®^ product and sildenafil-containing tablets obtained from illegal sources (the darknet). Specifically, the analyzed material consisted of samples seized by Polish law enforcement authorities from unverified vendors operating within the Central European darknet market. The study utilized thermal methods, specifically Thermogravimetry (TG), Derivative Thermogravimetry (DTG), and calculated Differential Thermal Analysis (c-DTA), as well as colorimetric analysis based on the International Commission on Illumination (CIE) L*a*b* system. **Methods**: Thermal analyses enabled the assessment of the thermal stability of the tested samples, identification of characteristic stages of thermal decomposition, and determination of differences in thermal behavior between the pure substance, the original preparation, and darknet samples. In turn, color measurements in the CIE L*a*b* space allowed for an objective comparison of tablet appearance and determination of the degree of color similarity to the original product. **Results**: The obtained results showed that only a few samples (V1, V3, V4, V6, V8) exhibited features similar to the original Viagra^®^, both in terms of thermal profile and color. Most of the tested tablets were characterized by significant variability in physicochemical properties, indicating a lack of quality control and inconsistency in formulation. Samples V2 and V7 deviated particularly strongly—both thermally and visually—suggesting that they might not contain the original active substance or contained it in a different chemical form. **Conclusions**: The use of combined thermal and colorimetric methods proved to be an effective tool in the identification of counterfeit pharmaceutical products, enabling simultaneous evaluation of their composition and authenticity. The results confirm the validity of employing integrated physicochemical analyses for the detection of falsified medicines present on the illegal market.

## 1. Introduction

The counterfeiting of drugs is currently one of the most serious threats to global public health. The dynamic development of online commerce, including the rise in the Darknet—a hidden part of the Internet accessible only through specialized software and not indexed by standard search engines—has created favorable conditions for the uncontrolled distribution of pharmacologically active substances. Among the most frequently counterfeited medicines are those used to treat erectile dysfunction, particularly products containing sildenafil, a phosphodiesterase type 5 inhibitor (PDE-5 inhibitor) [1,2,3]. Drugs containing sildenafil are a particularly attractive target for counterfeiters due to their high demand, ease of distribution, and the social stigma associated with their purchase. As a result, many consumers opt to buy these products online, often from unverified sources [3,4].

Studies conducted by the World Health Organization (WHO) and the European Medicines Agency (EMA) have shown that a significant portion of such preparations available online—especially on Darknet platforms—do not contain the declared amount of the active ingredient, are contaminated, or contain other chemical substances of unknown pharmacological activity [5,6,7,8]. The use of unverified products containing sildenafil can lead to serious adverse effects, including cardiovascular disorders, drug interactions (e.g., with nitrates), and poisoning with toxic substances [9,10,11]. The lack of quality control, improper manufacturing processes, and unknown origin of raw materials make these counterfeit products a real threat to human health and life [12,13].

The growing scale of drug counterfeiting necessitates the development of rapid, precise, and portable analytical methods that allow for the reliable identification of counterfeit pharmaceutical products [14,15]. Among the analytical techniques used in quality control and authenticity verification of sildenafil-containing preparations, the following can be distinguished: Liquid Chromatography (HPLC, UPLC) coupled with UV detection or mass spectrometry (LC-MS/MS), enabling quantitative determination of active substances and detection of impurities [16,17,18]. Infrared (FT-IR, NIR) and Raman spectroscopy, which allow for rapid, non-destructive comparison of spectral fingerprints between reference and tested samples [19,20]. X-ray powder diffraction (XRPD), used to analyze crystal structures that may differ between authentic and counterfeit products [21,22].

Beyond these established methods, recent advancements have introduced sophisticated imaging and spectroscopic tools for authenticity assessment. Jung et al. developed a non-destructive methodology based on image processing and statistical analysis; using a Video Spectral Comparator (VSC), they modeled the RGB color components of authentic tablets and employed the Bhattacharyya distance to quantify the adherence of test samples to the authentic distribution [23]. Furthermore, Harwacki et al. demonstrated the efficacy of solid-state nuclear magnetic resonance (ssNMR) spectroscopy, specifically ^12^C CPMAS, to identify non-pharmacopoeial cellulose in counterfeit formulations. This detailed excipient analysis facilitated the development of a rapid, low-cost chemical dyeing technique capable of distinguishing counterfeit products based on the reaction of cellulosic fillers to specific staining reagents [24].

In recent years, thermogravimetric methods such as: thermogravimetric analysis (TGA), different thermal analysis (DTA) and differential scanning calorimetry (DSC) have gained increasing importance in the study of drug authenticity, including sildenafil-based preparations [21,25]. These methods measure changes in a sample’s mass (TGA) or heat flow (DSC) as a function of temperature. This allows researchers to determine the thermal stability, melting temperature, decomposition points, and phase transition profiles of both active pharmaceutical ingredients (APIs) and excipients.

The advantages of thermal analysis include: low-sample requirement—only a small amount of material is needed to obtain a complete thermal profile; speed of analysis—results can be obtained within minutes, without the need for sample dissolution; high sensitivity and reproducibility—even small differences in composition, purity, or crystalline form result in detectable changes in TGA/DTA/DSC curves; ability to distinguish authentic from counterfeit products—counterfeit formulations often contain different excipients, leading to distinct thermal behaviors; utility in comparative analysis—TGA, DTA and DSC data can be compared with reference products to confirm or reject authenticity [26].

Colorimetric analysis serves as a vital complementary tool for rapid screening. Specifically, the CIE L*a*b* system, developed by the International Commission on Illumination, provides a standardized, objective method for quantifying color perception. In this three-dimensional model, L* represents lightness (0 for black to 100 for white), a* denotes the position on the green–red axis, and b* denotes the position on the blue–yellow axis. By calculating the total color difference (ΔE*) between a suspect sample and a reference standard, analysts can detect subtle deviations in tablet coating and pigmentation that are often invisible to the naked eye, thereby flagging potential counterfeits based on visual inconsistency [27,28].

The increasing quality of falsified medicines necessitates a multi-modal analytical approach. Relying solely on visual inspection is insufficient due to improved imitation of packaging and coating, while standalone advanced chemical analysis (e.g., HPLC-MS) can be time-consuming and expensive for routine screening. Therefore, combining rapid objective colorimetry with thermal profiling provides a dual-layer verification process, allowing for the simultaneous assessment of both visual consistency and physicochemical formulation stability.

The aim of this study was to conduct a comparative analysis of the original Viagra^®^ preparation and sildenafil-containing drugs obtained from illegal sources (the darknet), using thermal methods (TG, DTG, c-DTA) and colorimetric methods (CIE L*a*b* system).

## 2. Results and Discussion

### 2.1. Thermal Analysis

#### 2.1.1. Thermogravimetric Analysis (TG)

Thermogravimetric analysis (TG) allows determining the temperatures at which the tested samples lose mass, i.e., undergo thermal decomposition. For pure sildenafil citrate, the process began at approximately 197 °C and continued up to around 229 °C, with a total mass loss of as much as 77% (Figure 1, Table 1). This result indicates complete decomposition of the active substance at that temperature, without the presence of stabilizers or excipients. The result is consistent with literature data [29,30].

The original Viagra^®^ tablet exhibited a different behavior—decomposition started earlier, at around 189 °C, and continued up to 369 °C, indicating a much wider degradation range and a smaller total mass loss (approximately 49%). Such behavior is typical of composite pharmaceutical formulations containing excipients that stabilize the material (Figure 1, Table 1) [26,31].

Samples obtained from the darknet showed highly variable behavior. The most similar to the original were V1 and V2, in which the onset of decomposition occurred between 189–191 °C, and the end around 354–369 °C (Figure 1, Table 1). However, their total mass loss was higher (about 60–65%), suggesting a lower content of stabilizing substances or a higher proportion of volatile components.

The remaining samples (V3 to V8) exhibited significantly lower total mass losses—between 27% and 41%—and less consistent temperature ranges (Figure 1, Table 1). This indicates a heterogeneous chemical composition, the presence of large amounts of thermally resistant fillers (e.g., starch, lactose, cellulose), or a lower amount of sildenafil [31,32,33].

The most distinct sample was V8, for which decomposition began only at around 313 °C, much higher than for the other samples (Figure 1, Table 1). This difference suggests that the sample may contain a different active ingredient or a chemically distinct form of sildenafil [26,31,32,33].

In summary, the TG analysis indicates that V1 and V2 may have contained sildenafil in an amount and form similar to the original product, whereas V3–V8 differed substantially in composition, with V8 likely representing an entirely different formulation.

#### 2.1.2. Analysis of the First Derivative of the TG Curve (DTG)

The DTG analysis shows the rate of mass loss as a function of temperature, allowing for a more detailed identification of individual degradation stages.

For pure sildenafil citrate, three characteristic stages can be distinguished: the first at approximately 54 °C (moisture loss), the second around 202 °C (onset of chemical structure decomposition), and the third near 320 °C (final molecular breakdown) (Figure 2, Table 2) [29,30].

The original Viagra^®^ exhibited four degradation stages, which is typical for complex drug formulations. The first stage appeared at approximately 60 °C, followed by peaks at 195 °C, 320 °C, and finally 356 °C. The decomposition rate values (%/min) were moderate, indicating good stability and a controlled thermal process (Figure 2, Table 2) [26].

The DTG curve of the core of the original Viagra^®^ tablet shows four thermal decomposition stages, including an additional peak at 356.6 °C, which is absent in the case of pure sildenafil citrate. The presence of this stage can be attributed to the decomposition of excipients present in the tablet formulation (Figure 3, Table 3). In particular, microcrystalline cellulose exhibits an intense DTG peak at around 359 °C, while magnesium stearate decomposes at 371.8 °C. The proximity of the 356.6 °C peak to these characteristic temperatures suggests that the additional decomposition stage of the tablet core originates mainly from microcrystalline cellulose, with a possible contribution from magnesium stearate. The first decomposition stage of the original Viagra^®^ tablet core (peak at 59.7 °C) is likely related to physical and chemical processes occurring in excipients with lower thermal stability. Within this temperature range, the first decomposition steps are observed for microcrystalline cellulose (55.4 °C) and croscarmellose sodium (70.8 °C). This suggests that the first DTG peak of the tablet core results from overlapping degradation effects of these two components together with the pure API. The first temperature peak of magnesium stearate (109.2 °C) was not recorded for the original Viagra^®^ sample, which is likely due to the very small amount of this excipient in the formulation—typically up to 1% [34,35].

In the case of darknet samples, formulations V1 and V2 showed DTG profiles very similar to the original Viagra^®^, with comparable degradation temperatures but lower peak intensities, suggesting a lower content or partial dilution of sildenafil.

Samples V3–V8 displayed a more complex degradation pattern, with additional decomposition stages at lower temperatures (120–130 °C) not present in the original Viagra^®^ (Figure 2, Table 2). This indicates the presence of other organic components, such as inexpensive fillers or sugars.

Sample V8 exhibited a distinctly different profile, with the first decomposition stage only at 310 °C, which does not correspond to any legal sildenafil formulation (Figure 2, Table 2).

Overall, DTG analysis reveals that only samples V1 and V2 show degradation profiles similar to the original Viagra^®^, whereas the remaining samples represent mixed, non-reproducible compositions typical of uncontrolled production processes.

#### 2.1.3. Differential Thermal Analysis (c-DTA)

The c-DTA allows observation of the thermal effects of ongoing reactions—endothermic (heat-absorbing, e.g., melting) and exothermic (heat-releasing, e.g., chemical decomposition) processes [26,31,36].

Pure sildenafil citrate exhibited three distinct thermal effects. The first endothermic one at approximately 195–202 °C (melting), the second exothermic at around 224 °C (onset of chemical decomposition), and the third exothermic effect between 331–355 °C, corresponding to the final molecular breakdown (Figure 4a, Table 4) [37].

The original Viagra^®^ showed a very similar thermal profile; however, the areas of the thermal effects were smaller, which can be attributed to the presence of excipients—fillers and binders that partially absorb heat and stabilize the material. In addition, a slight shift in the third peak toward higher temperatures was observed, which is associated with the presence of microcrystalline cellulose, known to decompose at higher temperatures (Figure 4a, Table 4) [31].

The darknet samples were much more diverse. Samples V1 and V2 resembled the thermal profile of the original Viagra^®^, but exhibited additional exothermic peaks in the range of 265–300 °C, indicating the presence of other organic substances that undergo combustion or decomposition within this temperature range (Figure 4a, Table 4).

Samples V3–V7 showed strong endothermic effects already at around 110–130 °C, which were absent in the original Viagra^®^. This indicates the presence of other fillers or components with lower melting points, such as sugars. For many of these samples, the exothermic effects were also significantly weaker, which may suggest a lower amount of active substance (Figure 4a,b, Table 4).

Sample V9 exhibited a very simplified thermal pattern, with three thermal effects occurring in entirely different temperature ranges. This suggests that its composition is fundamentally different from all other samples, and that the sildenafil content—if present at all—is likely negligible (Figure 4b, Table 4).

### 2.2. Colorimetric Analysis

The color of pharmaceutical tablets is not only a matter of aesthetics but also an important indicator of quality, production control, and authenticity. The original Viagra^®^ has a characteristic light turquoise-blue shade resulting from a strictly controlled coloring process and the use of a specific pigment by the manufacturer, Pfizer.

In contrast, the darknet samples showed significant color variability compared with the authentic product. This variation results from differences in formulation, pigment quality, or in some cases, the complete absence of colorants.

The original Viagra^®^ tablet (V0) displays a medium brightness level and a balanced hue between blue and slightly greenish tones. It is not an intense blue but rather a soft, pastel turquoise with a calm and slightly matte appearance (Table 5). This recognizable and distinctive color differentiates Viagra^®^ from other tablets and serves as a hallmark of authenticity.

Sample V1 is almost indistinguishable from the original Viagra^®^. It shows nearly the same brightness and a very similar turquoise-blue hue. It may be slightly lighter and warmer in tone, but the differences are minimal and hardly perceptible to the naked eye (Table 5). This suggests that the producers of V1 attempted to closely reproduce the original color to enhance product credibility. Visually, V1 could be considered a very good imitation of authentic Viagra^®^.

Sample V2, on the other hand, bears no resemblance to the original. Its color is dark, matte, and warm brown-beige, completely lacking the characteristic blue tint (Table 5). Instead of a cool turquoise tone, the tablet has an earthy, yellowish-red appearance, which may result from the use of a different pigment or the absence of colorant, revealing the natural hue of the tablet mass. Such a distinct color difference strongly suggests that V2 is unrelated to the original product and was likely made from entirely different ingredients.

Sample V3 is somewhat similar to the original but slightly darker and cooler in tone. Its color shifts toward a greenish-blue shade, giving it a colder and less vivid appearance (Table 5). The difference is subtle and remains within the range of visual similarity; therefore, V3 can be considered a moderately successful attempt to replicate the color of Viagra^®^.

The color of sample V4 is almost identical to the original but slightly more subdued. It lacks the same brightness and may appear a bit more “grayish” or “muted” (Table 5). This impression may result from a different coating composition or the use of a less pure pigment. Overall, V4 remains within the characteristic color range of Viagra^®^ and could also visually mislead consumers.

Sample V5 shows a distinctly lighter and more vivid color than the original. It is a bright turquoise with higher color saturation, and the tablet surface appears glossier and optically cleaner (Table 5). Although this color may seem more attractive, it clearly differs from the authentic product. The impression suggests that a different or more concentrated pigment was used. The bright, eye-catching tone makes the tablet appear “newer,” but it is not identical to Viagra^®^.

Sample V6, like V1, is almost indistinguishable from the authentic Viagra^®^ tablet. It exhibits a very similar hue, brightness, and saturation (Table 5). Compared with the original, it may be only slightly lighter and more bluish, but the difference becomes noticeable only under direct laboratory lighting. V6 therefore represents an excellent visual imitation of the original color.

Sample V7 clearly differs from all others. Its color is dark, deep, and highly saturated, resembling navy blue or purplish-blue. The tone is cool, intense, and entirely unlike the soft turquoise of the original (Table 5). The tablet appears artificial, suggesting the use of a cheap, strong food-grade dye with a completely different spectral profile. V7 can thus be identified as a visually distinct product, strongly implying a different source of origin and a lack of quality control.

Sample V8 is very similar to the original, though its hue appears slightly more greenish and cool. The brightness and saturation are nearly identical, but the shift toward a cooler blue tone gives the tablet a slightly different appearance in daylight—more of a sea-blue shade (Table 5). This difference is subtle but noticeable. V8 can be regarded as a good visual replica, although its hue suggests a different ratio or type of coloring agent.

### 2.3. Study Limitaions and Future Plans

Although the combined thermal and colorimetric analysis proved effective in distinguishing authentic from counterfeit Viagra^®^ tablets, the present study has certain limitations, primarily related to the relatively small sample size and the lack of full chemical identification of the non-standard excipients detected. While thermogravimetric methods precisely revealed significant differences in thermal stability and decomposition profiles, they do not allow for the definitive structural characterization of the specific fillers responsible for these deviations. Similarly, colorimetric measurements in the CIE L*a*b* system, being point-based averages, effectively quantified general hue differences but may have overlooked subtle variations in surface texture and coating homogeneity that are critical for forensic analysis. To address these limitations, future research will focus on expanding the dataset to include a wider array of falsified samples from diverse illicit distribution channels, thereby enabling the construction of a more robust statistical model of counterfeit variability. It should be noted that while thermal and colorimetric analyses effectively detect formulation inconsistencies and differences in excipient composition, they serve primarily as rapid screening tools. For a complete quality verification, particularly regarding the precise quantification of the active substance dose, these methods should ideally be complemented by standard pharmacopoeial techniques such as HPLC.

A pivotal expansion of our methodology will involve the integration of solid-state nuclear magnetic resonance (ssNMR) spectroscopy, as demonstrated by Harwacki et al., to generate detailed molecular fingerprints of the tablet matrix. This advanced spectroscopic technique will facilitate the precise identification of non-pharmacopoeial excipients, such as industrial-grade cellulose, whose presence was only indirectly suggested by the altered thermal decomposition stages observed in our current results. Furthermore, we intend to implement the computerized image processing workflow proposed by Jung et al., utilizing the Video Spectral Comparator and the Bhattacharyya distance to objectively assess visual consistency and detect micro-texture anomalies that elude standard colorimetry. We also plan to explore the correlation between the observed thermal instability and the reactivity of tablet components to specific chemical staining reagents, aiming to develop rapid, low-cost field tests that bridge the gap between complex instrumental analysis and frontline screening. By combining these high-resolution structural insights with simple colorimetric and thermal data, we aim to establish a multi-tiered detection protocol that is both scientifically rigorous and practically deployable. Additionally, future studies will incorporate toxicological assessments of the degradation products identified in thermally unstable samples to better quantify the potential health risks posed to consumers. We will also investigate the kinetics of sildenafil degradation under uncontrolled storage conditions typical of the black market, determining how the variable matrix composition influences the shelf-life of these illicit products. Ultimately, our goal is to synthesize these thermal, spectroscopic, and visual data into a comprehensive digital library that can serve as a reference standard for law enforcement and regulatory bodies. Such an integrated diagnostic algorithm would significantly enhance the capability to intercept counterfeit pharmaceuticals at various points in the supply chain, ensuring public safety through more reliable authenticity verification.

## 3. Materials and Methods

### 3.1. Tested Samples

The study utilized pure sildenafil citrate (Sigma-Aldrich, Merck, Darmstadt, Germany), samples of Viagra^®^ 100 mg tablets (Pfizer, New York, NY, USA) (V0) obtained from a legal source (a licensed pharmacy), as well as tablet samples resembling the Viagra^®^ preparation (V1–V8) acquired from unauthorized sources (purchased online). The falsified medicinal products analyzed in this study were obtained directly from Polish law enforcement authorities and judicial bodies. These samples were released for scientific investigation only after the formal conclusion of relevant criminal proceedings and evidentiary protocols. Consequently, the materials were no longer subject to legal retention requirements at the time of acquisition. Following the completion of all analytical assessments, the remaining samples were processed for final elimination. All materials were disposed of in strict accordance with established safety regulations and standard operating procedures for the destruction of pharmaceutical and chemical waste.

### 3.2. Thermal Analyses (TGA, c-DTA)

#### 3.2.1. Sample Preparation for Thermal Testing

Samples of original Viagra^®^ 100 mg tablets (V0) and preparations purchased online (V1–V8) were cleaned of their coating layer and ground into a homogeneous powder using a mortar and pestle. The prepared powder mass was then used for thermogravimetric measurements. A reference sample of sildenafil citrate was purchased from Sigma-Aldrich (Merck). The sample was in powder form with a purity of >99.8%.

#### 3.2.2. Measurement Apparatus

The analyses were performed using a TG 209 F3 Tarsus thermogravimeter (Netzsch, Selb, Germany), which allows simultaneous measurements with differential thermal analysis (c-DTA).

Measurement parameters: temperature range from 35 °C to 600 °C; heating rate of 10 K/min; measurements carried out in a nitrogen (N_2_) atmosphere with a flow rate of 40 mL/min. The mass of the tested samples was 10 mg. An alumina (Al_2_O_3_) crucible was used for the measurements.

For calibration of the c-DTA curve, high-purity standards (indium, zinc, tin, aluminum, bismuth, silver) were used in accordance with the manufacturer’s recommendations.

#### 3.2.3. Analysis Procedure

Each sample was weighed using an analytical balance with an accuracy of ±0.01 mg and placed in a measuring crucible. The samples were heated within the specified temperature range under a controlled flow of inert gas. During the analysis, the following parameters were recorded: the change in sample mass as a function of temperature (TGA), the first derivative of the TG curve (DTG), and the thermal effects accompanying physical or chemical transformations (c-DTA). Each measurement was repeated three times (n = 3) under identical conditions to ensure the reproducibility of the results. The TG, DTG, and c-DTA curves presented in this study are representative of the obtained datasets.

For each sample, TGA, DTG, and c-DTA curves were generated and subsequently analyzed using Proteus 8.0 software (Netzsch, Selb, Germany).

### 3.3. Colorimetric Analysis

#### 3.3.1. Sample Preparation for Colorimetric Testing

Reference samples—Viagra^®^ 100 mg tablets (Pfizer) purchased from a licensed pharmacy—and preparations with a declared sildenafil content obtained from unauthorized sources (Internet) were labeled with laboratory codes from V0 to V8.

The tablets were not damaged; their surfaces were analyzed in an intact state (the external colored coating layer).

#### 3.3.2. Equipment and Measurement Conditions

Reference Color measurements were performed using an NH310 reflective colorimeter (3nh, Guangzhou, China). The measurement parameters were as follows: measurement geometry- d/8° (integrating sphere); instrument calibration performed using a standard white calibration plate (Y = 93.7, x = 0.3164, y = 0.3330); aperture diameter- 8 mm; ambient temperature and humidity during measurement- 22 °C [±2 °C], RH = 45% [±5%].

Each measurement was repeated six times at different points on the tablet surface, and the arithmetic mean values of the L*, a*, and b* parameters were then calculated.

#### 3.3.3. Analyzed Color Parameters

The color of each sample was described in the 3D CIE L*a*b* color space: L*—lightness (0 = black, 100 = white); a*—color component along the red–green axis (positive values = red, negative values = green); b*—color component along the yellow–blue axis (positive values = yellow, negative values = blue) [27,28].

Based on the a* and b* values, the following additional parameters were calculated (1), (2) [28]:

**C*** (chroma, color saturation) [28]:(1)$$C^{*}=\sqrt{{\left(a^{*}\right)}^{2}+{\left(b^{*}\right)}^{2}}$$

**h°** (hue angle) [28]:(2)$$h{}^{\circ}=\arctan\left(\frac{b^{*}}{a^{*}}\right)$$

Color differences (**ΔE***) between the tested sample and the original were calculated using the Formula (3) [28]:(3)$$\Delta E^{*}=\sqrt{{\left(\Delta L^{*}\right)}^{2}+{\left(\Delta a^{*}\right)}^{2}+{\left(\Delta b^{*}\right)}^{2}}$$

#### 3.3.4. Data Processing and Statistical Analysis

For each sample, the values of L*, a*, b*, C*, and h° were compiled, and the mean color difference (ΔE*) relative to the original sample was calculated.

Criteria for interpreting color differences using the ΔE* parameter [28]:ΔE* < 0.5—difference imperceptible to the human eye,0.5 ≤ ΔE* < 1.5—difference barely visible,1.5 ≤ ΔE* < 3.0—difference poorly visible,3.0 ≤ ΔE* < 6.0—difference visible,ΔE* ≥ 6.0—difference very visible.

The results were statistically analyzed using Statistica 13.0 software (TIBCO Software Inc., Palo Alto, CA, USA).

A one-way ANOVA (*p* < 0.05) was applied to assess the significance of differences between the L*, a*, and b* values for the original and tested samples.

## 4. Conclusions

Combating the phenomenon of drug counterfeiting, particularly of sildenafil-containing preparations sold on the darknet, requires a comprehensive approach that integrates legislative, educational, and technological measures. The development of rapid, automated analytical methods—including thermogravimetry and colorimetry—is a key component of public health protection strategies. The complementary use of different physical and thermal techniques enables precise identification of falsified products and contributes to reducing their circulation on the market.

Analysis of all applied thermal methods (TG, DTG, and c-DTA) showed that the original Viagra^®^ exhibits greater thermal stability, resulting from the presence of excipients that enhance the stability of the drug.

Among the online-purchased samples, V1 and V2 displayed thermal profiles most similar to that of the original Viagra^®^. It can be assumed that they contain sildenafil in the correct chemical form, although not necessarily in the full therapeutic dose.

Samples V3–V7 differed substantially in composition from the reference product. They contained various accompanying substances, lower amounts of sildenafil, and their thermal characteristics indicated poor quality or uncontrolled manufacturing processes.

Sample V8 deviated entirely from the reference standard. Its thermal properties suggest the absence of the authentic active ingredient or substitution with another, unidentified compound.

Colorimetric analysis likewise demonstrated that only a few darknet samples (V1, V3, V4, V6, V8) visually resembled the original Viagra^®^. Most of the online-obtained samples exhibited significant color variability, confirming the lack of quality control and formulation consistency.

Drastic differences, such as those observed in samples V2 and V7, indicate that some “Viagra” tablets from illegal sources may not contain the genuine active substance at all, but rather a random mixture of components. In practice, this means that the color of darknet tablets is not a reliable indicator of authenticity but instead reflects the absence of standardization—a typical feature of counterfeit products manufactured outside pharmaceutical supervision.

Based on the obtained results, we propose the following standard parameters for discriminating authentic Viagra^®^ tablets using the combined method: (1) a colorimetric difference in ΔE* < 1.5 relative to the reference; (2) a total mass loss in TG analysis within the narrow range of 48–50%; and (3) the presence of a specific DTG decomposition peak at approximately 356 °C. Samples failing to meet these simultaneous criteria (e.g., V2–V8) should be classified as potential counterfeits.

## Figures and Tables

**Figure 1 pharmaceuticals-19-00078-f001:**
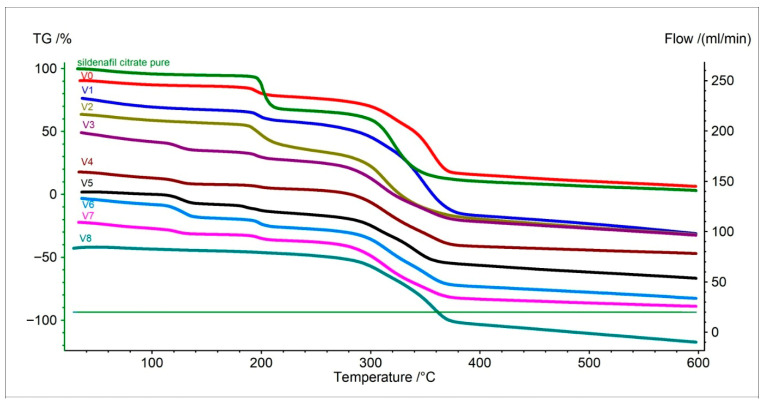
Thermogravimetric (TG) curves of pure API (sildenafil citrate), original Viagra^®^ (V0), and Viagra tablets purchased online (V1–V8).

**Figure 2 pharmaceuticals-19-00078-f002:**
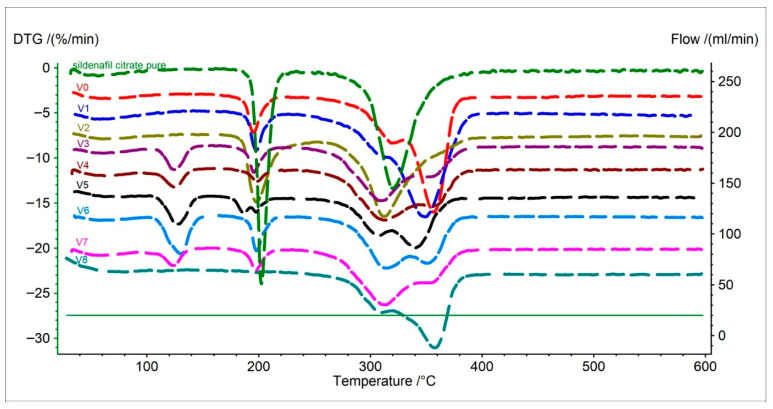
First derivative thermogravimetric (DTG) curves of pure API (sildenafil citrate), original Viagra^®^ (V0), and Viagra tablets purchased online (V1–V8).

**Figure 3 pharmaceuticals-19-00078-f003:**
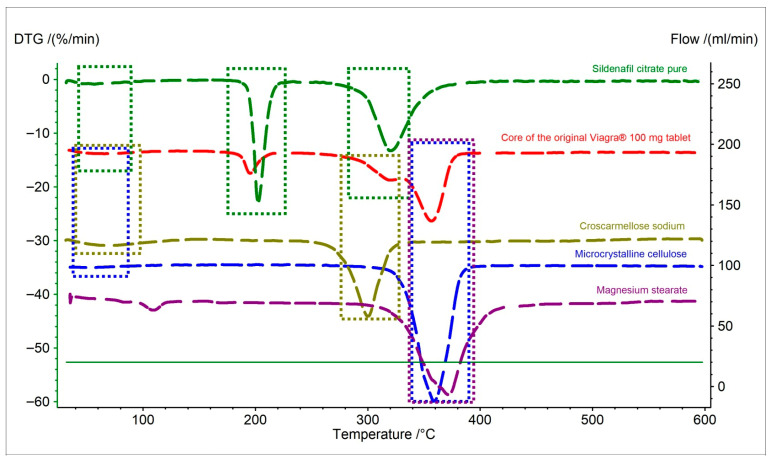
First derivative thermogravimetric (DTG) curves of pure API (sildenafil citrate), the core of the original Viagra^®^ tablet, and excipients (microcrystalline cellulose, croscarmellose sodium, and magnesium stearate). The frame indicates the decomposition stages of the original Viagra^®^ derived from pure sildenafil citrate and individual excipients.

**Figure 4 pharmaceuticals-19-00078-f004:**
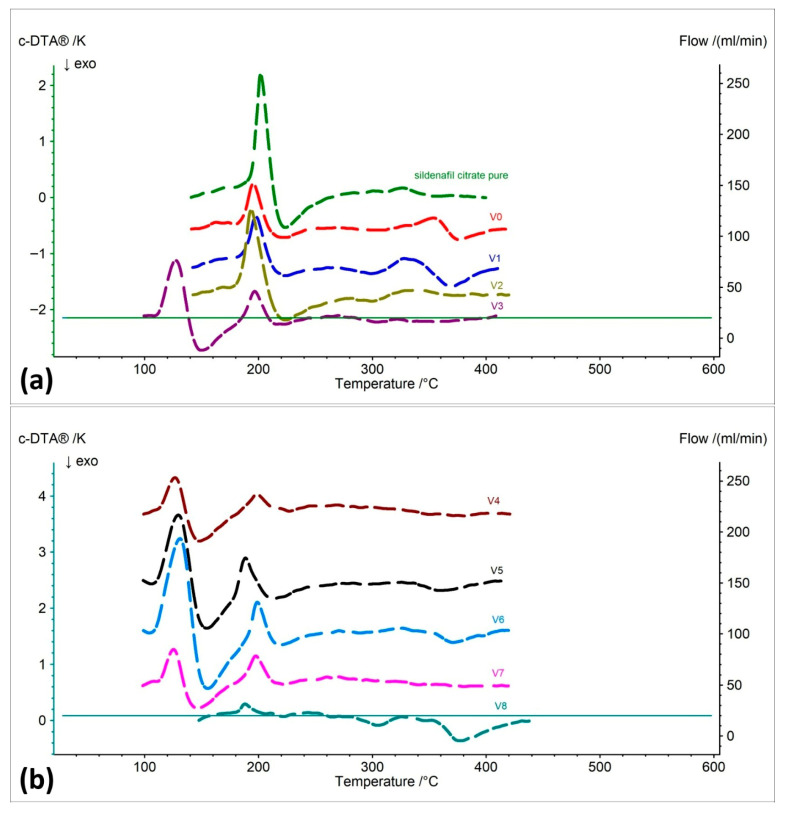
Calculated differential thermal analysis (c-DTA) curves of (**a**) pure API (sildenafil citrate), original Viagra^®^ (V0), and Viagra tablets purchased online (V1–V3), and (**b**) Viagra tablets purchased online (V4–V8).

**Table 1 pharmaceuticals-19-00078-t001:** Parameters obtained from thermogravimetric (TG) curves for pure API (sildenafil citrate), original Viagra^®^ (V0), and Viagra tablets purchased online (V1–V8).

Samples	Onset [°C]	Mid [°C]	Infection [°C]	End [°C]	Mass Change [%]
Sildenafil citrate pure	197	310	202.4	228.8	−77.05
V0	189.2	349.1	355.7	369.1	−48.86
V1	191.4	342.1	350.1	369.3	−59.74
V2	189	305.4	311.4	354.6	−64.64
V3	190.3	318.3	308.7	352.2	−37.82
V4	190.5	323.2	313.2	362.7	−40.29
V5	179.6	332.4	339.5	360.1	−27.78
V6	192.8	332.9	312.3	360.5	−33.08
V7	192.2	319.5	314.1	357.1	−40.88
V8	313.5	335.2	359	371.4	−48.71

**Table 2 pharmaceuticals-19-00078-t002:** Parameters obtained from first derivative thermogravimetric (DTG) curves for pure API (sildenafil citrate), original Viagra^®^ (V0), and Viagra tablets purchased online (V1–V8).

Samples	STAGE 1		STAGE 2		STAGE 3		STAGE 4		STAGE 5		STAGE 6	
	[°C]	%/min	[°C]	%/min	[°C]	%/min	[°C]	%/min	[°C]	%/min	[°C]	%/min
Sildenafil citrate pure	53.9	−0.84	202.5	−23.94	319.9	−13.37	-	-	-	-	-	-
V0	59.7	−0.67	195.5	−4.40	319.5	−5.61	356.6	−13.26	-	-	-	-
V1	58.1	−1.23	198.0	−4.68	313.1	−5.42	349.4	−12.08	-	-	-	-
V2	60.5	−0.87	198.0	−7.86	312.5	−9.46	-	-	-	-	-	-
V3	61.0	−1.24	125.1	−3.05	195.7	−3.35	308.7	−6.50	353.0	−3.86	-	-
V4	62.9	−0.95	125.1	−2.19	198.6	−1.56	312.9	−5.86	355.0	−4.32	-	-
V5	62.8	−0.42	127.0	−3.44	186.7	−2.22	200.6	−1.72	307.2	−4.77	339.9	−6.09
V6	57.6	−0.84	130.5	−4.56	198.3	−4.23	314.5	−6.14	351.0	−5.62	-	-
V7	58.6	−0.96	123.7	−2.09	197.2	−2.69	313.5	−6.43	353.0	−4.00	-	-
V8	65.9	−0.38	310.3	−4.93	357.9	−8.83	-	-	-	-	-	-

**Table 3 pharmaceuticals-19-00078-t003:** Parameters obtained from first derivative thermogravimetric (DTG) curves for pure API (sildenafil citrate), the core of the original Viagra^®^ tablet, and excipients (microcrystalline cellulose, croscarmellose sodium, and magnesium stearate). The steps related to the decomposition of the core of the original Viagra^®^ tablet are highlighted in bold.

Samples	Number of Thermal Decomposition Stages	[°C] (%/min)	[°C] (%/min)	[°C] (%/min)	[°C] (%/min)	[°C] (%/min)
Sildenafil citrate pure	3	53.9 (−0.84)	-	202.5 (−23.94)	319.9 (−13.37)	-
Core of the original Viagra^®^ tablet	**4**	**59.7** **(−0.67)**	**-**	**195.5** **(−4.40)**	**319.5** **(−5.61)**	**356.6** **(−13.26)**
Microcrystalline cellulose	2	55.4 (−0.76)	-	-	-	359.0 (−25.65)
Croscarmellose sodium	2	70.8 (−1.30)	-	-	300.2 (−14.47)	-
Magnesium stearate	2	-	109.2 (−1.89)	-	-	371.8 (−17.61)

**Table 4 pharmaceuticals-19-00078-t004:** Parameters obtained from calculated differential thermal analysis (c-DTA) curves for pure API (sildenafil citrate), original Viagra^®^ (V0), and Viagra tablets purchased online (V1–V8). The stage associated with the melting point is highlighted in bold.

Samples	Stage	Onset [°C]	Peak [°C]	Area [K*s]	Reaction
Sildenafil citrate pure	**1**	**194.5**	**201.9**	**108.41**	**ENDOTHERMIC**
	2	-	223.6	138.06	EXOTHERMIC
	3	331.6	355.4	17.88	EXOTHERMIC
V0	**1**	**186.8**	**195.3**	**52.68**	**ENDOTHERMIC**
	2	-	222.7	19.98	EXOTHERMIC
	3	358.9	376.5	56.86	EXOTHERMIC
V1	**1**	**188.4**	**197.6**	**41.92**	**ENDOTHERMIC**
	2	-	223.5	31.92	EXOTHERMIC
	3	264.9	298.8	37.91	EXOTHERMIC
	4	333.1	369.8	93.12	EXOTHERMIC
V2	**1**	**185.1**	**193.5**	**85.94**	**ENDOTHERMIC**
	2	-	223.8	88.20	EXOTHERMIC
	3	279.6	299.5	14.29	EXOTHERMIC
	4	349.6	360.4	7.84	EXOTHERMIC
V3	1	113.0	127.2	102.33	ENDOTHERMIC
	**2**	**188.2**	**196.6**	**31.97**	**ENDOTHERMIC**
	3	-	222.3	12.72	EXOTHERMIC
	4	277.6	304.0	7.56	EXOTHERMIC
	5	328.3	333.3	6.33	EXOTHERMIC
V4	1	114.2	126.6	51.31	ENDOTHERMIC
	**2**	**192.9**	**198.6**	**14.63**	**ENDOTHERMIC**
	3	-	227.0	6.55	EXOTHERMIC
	4	296.5	350.7	19.83	EXOTHERMIC
V5	1	111.7	129.6	145.60	ENDOTHERMIC
	**2**	**179.8**	**188.5**	**44.05**	**ENDOTHERMIC**
	3	-	212.9	21.83	EXOTHERMIC
	4	334.1	358.9	32.36	EXOTHERMIC
V6	1	111.3	131.0	204.82	ENDOTHERMIC
	**2**	**192.4**	**199.0**	**30.50**	**ENDOTHERMIC**
	3	-	219.2	19.21	EXOTHERMIC
	4	272.9	289.0	6.01	EXOTHERMIC
	5	355.5	370.9	30.87	EXOTHERMIC
V7	1	114.2	125.3	50.90	ENDOTHERMIC
	**2**	**188.7**	**197.6**	**26.15**	**ENDOTHERMIC**
	3	-	223.0	14.35	EXOTHERMIC
	4	329.3	340.9	3.30	EXOTHERMIC
V8	**1**	**182.6**	**188.2**	**8.30**	**ENDOTHERMIC**
	2	275.2	304.4	20.47	EXOTHERMIC
	3	362.4	376.8	74.62	EXOTHERMIC

**Table 5 pharmaceuticals-19-00078-t005:** Parameters obtained from colorimetry in the CIE L*a*b* system and the C*, h°, and ΔE* parameters for original Viagra^®^ (V0) and Viagra tablets purchased online (V1–V8). Results are presented as means ± SD (n = 6). Differences were considered statistically significant at *p* < 0.05.

Sample	Color Parameters	ΔE* [±SD]	Picture ofTablets
CIE L*a*b*	C* h°
L* [±SD]	a* [±SD]	b* [±SD]	C* [±SD]	h° [±SD]
V0	59.90 [±0.06]	−3.18 [±0.03]	4.18 [±0.06]	5.25 [±0.04]	127.28 [±0.56]	-	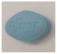
V1	60.74 [±0.01]	−2.20 [±0.01]	5.78 [±0.03]	6.19 [±0.03]	110.83 [±0.21]	2.05[±0.06]	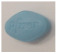
V2	30.97 [±0.03]	3.72 [±0.05]	9.65 [±0.05]	10.34 [±0.04]	68.92 [±0.29]	30.24[±0.06]	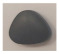
V3	57.60 [±0.01]	−4.28 [±0.03]	2.35 [±0.01]	4.88 [±0.03]	151.20 [±0.17]	3.14[±0.06]	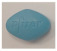
V4	57.23 [±0.01]	−3.43 [±0.02]	2.52 [±0.01]	4.26 [±0.02]	143.69 [±0.15]	3.15[±0.06]	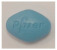
V5	65.23 [±0.01]	−2.72 [±0.05]	7.24 [±0.04]	7.73 [±0.05]	110.62 [±0.24]	6.16[±0.06]	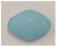
V6	61.57 [±0.01]	−2.96 [±0.04]	5.28 [±0.03]	6.05 [±0.02]	119.28 [±0.30]	2.01[±0.06]	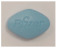
V7	46.20 [±0.01]	−5.46 [±0.02]	−7.47 [±0.03]	9.25 [±0.02]	233.82 [±0.06]	18.13[±0.06]	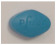
V8	61.98 [±0.01]	−5.50 [±0.01]	3.03 [±0.01]	6.28 [±0.01]	151.14 [±0.06]	3.32[±0.06]	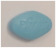

## Data Availability

The original contributions presented in this study are included in the article. Further inquiries can be directed to the corresponding author.

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
