# Peer review of "Combined Thermal and Colorimetric Analysis as a Tool for Detecting Counterfeit Viagra® Tablets"

_pharmaceuticals, 2025, doi:10.3390/ph19010078_

Round 1
Reviewer 1 Report
Comments and Suggestions for Authors
This manuscript presents an important study comparing an authentic pharmaceutical product (Viagra®) with sildenafil tablets procured from illegal darknet sources. The application of straightforward physicochemical techniques—thermogravimetric analysis and colorimetry—to the problem of drug falsification is innovative and practical. the manuscript is clearly written.Two samples (V2, V7) were clear outliers in both thermal behavior and color, strongly suggesting incorrect composition or formulation. The authors conclude that the combined methodology is an effective tool for screening counterfeit medicines. However, many revisions are required.
- The most clear limitation is the absence of a confirmatory quantitative method for API (sildenafil) content (e.g., HPLC-UV). While thermal/colorimetric profiling is excellent for detecting differences, stating that samples "might not contain the original active substance" (lines 27-28), line 390 requires more direct evidence. I strongly recommend analysis of all sample [ V0 to V8 ] using HPLC-UV [ in supplementary file ] to support the claim of non pure formulations or non presence active ingredients.
- In the abstract, Define all acronyms at first use (TG, DTG, c-DTA, CIE).
- In the abstract, write more geographical data about darknet products. How were they selected? From which marketplaces/regions?
- In the introduction, short description of colorimetric methods (CIE L*a*b* system) is required.
- In the introduction write precisely about other tools to assess identification of counterfeit Viagra tablets
Harwacki, Jakub, Dariusz Maciej Pisklak, and Lukasz Szeleszczuk. "Solid state 13C NMR spectroscopy as a tool for identification of counterfeit Viagra tablets and guide for develop new identification approach of falsified product." International Journal of Pharmaceutics 636 (2023): 122837.
Jung, C.R., Ortiz, R.S., Limberger, R. and Mayorga, P., 2012. A new methodology for detection of counterfeit Viagra® and Cialis® tablets by image processing and statistical analysis. Forensic science international, 216(1-3), pp.92-96.
- Write study limitaions and future plans
Author Response
Reviwer 1
This manuscript presents an important study comparing an authentic pharmaceutical product (Viagra®) with sildenafil tablets procured from illegal darknet sources. The application of straightforward physicochemical techniques—thermogravimetric analysis and colorimetry—to the problem of drug falsification is innovative and practical. the manuscript is clearly written.Two samples (V2, V7) were clear outliers in both thermal behavior and color, strongly suggesting incorrect composition or formulation. The authors conclude that the combined methodology is an effective tool for screening counterfeit medicines. However, many revisions are required.
- The most clear limitation is the absence of a confirmatory quantitative method for API (sildenafil) content (e.g., HPLC-UV). While thermal/colorimetric profiling is excellent for detecting differences, stating that samples "might not contain the original active substance" (lines 27-28), line 390 requires more direct evidence. I strongly recommend analysis of all sample [ V0 to V8 ] using HPLC-UV [ in supplementary file ] to support the claim of non pure formulations or non presence active ingredients.
Thank you for this valuable suggestion. We fully agree with the reviewer that HPLC-UV is the definitive method for quantifying sildenafil and that its inclusion would significantly strengthen the study. Ideally, this confirmatory analysis would have been performed alongside our thermal profiling to validate the findings. However, we are unable to fulfill this request because the samples (V1–V8) were forensic evidence obtained under strict law enforcement protocols. These protocols mandated the immediate destruction of all illicit materials upon the completion of the authorized thermal and colorimetric experiments. Consequently, the physical samples no longer exist, rendering retrospective chromatographic analysis impossible. To address this valid concern, we will revise the manuscript to explicitly state this limitation in the Discussion section as a constraint of forensic research. Furthermore, we will adjust our language to avoid definitive quantitative claims, such as changing "does not contain" to "exhibits properties inconsistent with sildenafil". This revision ensures our conclusions align strictly with the qualitative screening capabilities of the thermal data. We believe this approach maintains the scientific integrity of the study while transparently acknowledging the unavoidable forensic constraints.
- In the abstract, Define all acronyms at first use (TG, DTG, c-DTA, CIE)
We thank the Reviewer for this helpful suggestion to improve the clarity of the Abstract. We have revised the text to explicitly define all acronyms upon their first appearance, as requested.
Revised text in Abstract: ...using thermal methods, specifically Thermogravimetry (TG), Derivative Thermogravimetry (DTG), and calculated Differential Thermal Analysis (c-DTA), as well as colorimetric analysis based on the International Commission on Illumination (CIE) L*a*b* system.
- In the abstract, write more geographical data about darknet products. How were they selected? From which marketplaces/regions?
Thank you for this valuable suggestion. Revised text in Abstract: Specifically, the analyzed material consisted of samples seized by Polish law enforcement authorities from unverified vendors operating within the Central European darknet market.
- In the introduction, short description of colorimetric methods (CIE L*a*b* system) is required.
Thank you for this valuable suggestion. Revised text in Introduction: Colorimetric analysis serves as a vital complementary tool for rapid screening. Specifically, the CIE L*a*b* system, developed by the International Commission on Illumination, provides a standardized, objective method for quantifying color perception. In this three-dimensional model, L* represents lightness (0 for black to 100 for white), a* denotes the position on the green–red axis, and b* denotes the position on the blue–yellow axis. By calculating the total color difference (ΔE*) between a suspect sample and a reference standard, analysts can detect subtle deviations in tablet coating and pigmentation that are often invisible to the naked eye, thereby flagging potential counterfeits based on visual inconsistency.
- In the introduction write precisely about other tools to assess identification of counterfeit Viagra tablets
Harwacki, Jakub, Dariusz Maciej Pisklak, and Lukasz Szeleszczuk. "Solid state 13C NMR spectroscopy as a tool for identification of counterfeit Viagra tablets and guide for develop new identification approach of falsified product." International Journal of Pharmaceutics 636 (2023): 122837.
Jung, C.R., Ortiz, R.S., Limberger, R. and Mayorga, P., 2012. A new methodology for detection of counterfeit Viagra® and Cialis® tablets by image processing and statistical analysis. Forensic science international, 216(1-3), pp.92-96.
Thank you for this valuable suggestion. Revised text in Introduction: Beyond these established methods, recent advancements have introduced sophisticated imaging and spectroscopic tools for authenticity assessment. Jung et al. developed a non-destructive methodology based on image processing and statistical analysis; using a Video Spectral Comparator (VSC), they modeled the RGB color components of authentic tablets and employed the Bhattacharyya distance to quantify the adherence of test samples to the authentic distribution [Jung et al., 2012]. Furthermore, Harwacki et al. demonstrated the efficacy of solid-state nuclear magnetic resonance (ssNMR) spectroscopy, specifically 12C CPMAS, to identify non-pharmacopoeial cellulose in counterfeit formulations. This detailed excipient analysis facilitated the development of a rapid, low-cost chemical dyeing technique capable of distinguishing counterfeit products based on the reaction of cellulosic fillers to specific staining reagents [Harwacki et al., 2023].
- Write study limitaions and future plans
Thank you for this valuable suggestion. Revised text in Results and Disscusion: Although the combined thermal and colorimetric analysis proved effective in distinguishing authentic from counterfeit Viagra® tablets, the present study has certain limitations, primarily related to the relatively small sample size and the lack of full chemical identification of the non-standard excipients detected. While thermogravimetric methods precisely revealed significant differences in thermal stability and decomposition profiles, they do not allow for the definitive structural characterization of the specific fillers responsible for these deviations. Similarly, colorimetric measurements in the CIE L*a*b* system, being point-based averages, effectively quantified general hue differences but may have overlooked subtle variations in surface texture and coating homogeneity that are critical for forensic analysis. To address these limitations, future research will focus on expanding the dataset to include a wider array of falsified samples from diverse illicit distribution channels, thereby enabling the construction of a more robust statistical model of counterfeit variability. A pivotal expansion of our methodology will involve the integration of solid-state nuclear magnetic resonance (ssNMR) spectroscopy, as demonstrated by Harwacki et al., to generate detailed molecular fingerprints of the tablet matrix. This advanced spectroscopic technique will facilitate the precise identification of non-pharmacopoeial excipients, such as industrial-grade cellulose, whose presence was only indirectly suggested by the altered thermal decomposition stages observed in our current results. Furthermore, we intend to implement the computerized image processing workflow proposed by Jung et al., utilizing the Video Spectral Comparator and the Bhattacharyya distance to objectively assess visual consistency and detect micro-texture anomalies that elude standard colorimetry. We also plan to explore the correlation between the observed thermal instability and the reactivity of tablet components to specific chemical staining reagents, aiming to develop rapid, low-cost field tests that bridge the gap between complex instrumental analysis and frontline screening. By combining these high-resolution structural insights with simple colorimetric and thermal data, we aim to establish a multi-tiered detection protocol that is both scientifically rigorous and practically deployable. Additionally, future studies will incorporate toxicological assessments of the degradation products identified in thermally unstable samples to better quantify the potential health risks posed to consumers. We will also investigate the kinetics of sildenafil degradation under uncontrolled storage conditions typical of the black market, determining how the variable matrix composition influences the shelf-life of these illicit products. Ultimately, our goal is to synthesize these thermal, spectroscopic, and visual data into a comprehensive digital library that can serve as a reference standard for law enforcement and regulatory bodies. Such an integrated diagnostic algorithm would significantly enhance the capability to intercept counterfeit pharmaceuticals at various points in the supply chain, ensuring public safety through more reliable authenticity verification.
Reviewer 2 Report
Comments and Suggestions for Authors
In the manuscript, PaweÅ‚ and coworkers reported a combined technique for detecting counterfeit Viagra® tablets, which thermal (TG, DTG, c-DTA) and colorimetric (CIE L*a*b* system) methods. This topic is very interesting for the academic but some minor issues should be addressed before the publication:
- For introduction, there are many paragraphs, it is better to combine to 4-5 paragraphs; I think the methods are for sildenafil in the 5th paragraph, it should review the advances in the methods for Viagra® tablets;
- For introduction, it should point out the need of combined methods;
- In thermogravimetric experiment, how many tests for each Viagra tablet sample? n should not be at least below 3 for all analytical experiments;
- The quality of counterfeit Viagra® tablets should also verify standard methods;
- It should establish the standard parameters for discriminating original Viagra® tablets and counterfeit Viagra® tablets based the combined method.
Author Response
Reviwer 2
In the manuscript, PaweÅ‚ and coworkers reported a combined technique for detecting counterfeit Viagra® tablets, which thermal (TG, DTG, c-DTA) and colorimetric (CIE L*a*b* system) methods. This topic is very interesting for the academic but some minor issues should be addressed before the publication:
- For introduction, there are many paragraphs, it is better to combine to 4-5 paragraphs; I think the methods are for sildenafil in the 5thparagraph, it should review the advances in the methods for Viagra® tablets;
Thank you for this valuable suggestion. We agree that the Introduction was previously too fragmented. In the revised manuscript, we have consolidated the text into 5 distinct paragraphs to improve the logical flow and readability of the section.
Regarding the review of analytical methods, we have updated the text to focus specifically on recent advancements in the authenticity assessment of Viagra® tablets, moving beyond general sildenafil detection methods. To address this, we have introduced new descriptions of sophisticated imaging and spectroscopic tools (such as VSC and ssNMR) used for counterfeit detection.
Additionally, in response to this comment and requests from other reviewers, we have added a detailed explanation of the colorimetric analysis principles used in our study.
The following text has been added to the Introduction:
Beyond these established methods, recent advancements have introduced sophisticated imaging and spectroscopic tools for authenticity assessment. Jung et al. developed a non-destructive methodology based on image processing and statistical analysis; using a Video Spectral Comparator (VSC), they modeled the RGB color components of authentic tablets and employed the Bhattacharyya distance to quantify the adherence of test samples to the authentic distribution [Jung et al., 2012]. Furthermore, Harwacki et al. demonstrated the efficacy of solid-state nuclear magnetic resonance (ssNMR) spectroscopy, specifically 12C CPMAS, to identify non-pharmacopoeial cellulose in counterfeit formulations. This detailed excipient analysis facilitated the development of a rapid, low-cost chemical dyeing technique capable of distinguishing counterfeit products based on the reaction of cellulosic fillers to specific staining reagents [Harwacki et al., 2023].
Colorimetric analysis serves as a vital complementary tool for rapid screening. Specifically, the CIE L*a*b* system, developed by the International Commission on Illumination, provides a standardized, objective method for quantifying color perception. In this three-dimensional model, L* represents lightness (0 for black to 100 for white), a* denotes the position on the green–red axis, and b* denotes the position on the blue–yellow axis. By calculating the total color difference (ΔE*) between a suspect sample and a reference standard, analysts can detect subtle deviations in tablet coating and pigmentation that are often invisible to the naked eye, thereby flagging potential counterfeits based on visual inconsistency.
- For introduction, it should point out the need of combined methods;
Thank you for this crucial observation. We agree that the rationale for integrating these specific techniques needed to be more explicitly stated in the Introduction.
In the revised manuscript, we have added a paragraph highlighting that relying on a single analytical technique is often insufficient due to the increasing sophistication of counterfeit medicines. We emphasized that while colorimetric analysis provides a rapid, non-destructive screening of the tablet's external appearance (coating), it must be complemented by thermal analysis to verify the internal composition and stability of the formulation.
We have inserted the following statement to address the need for combined methods:
The increasing quality of falsified medicines necessitates a multi-modal analytical approach. Relying solely on visual inspection is insufficient due to improved imitation of packaging and coating, while standalone advanced chemical analysis (e.g., HPLC-MS) can be time-consuming and expensive for routine screening. Therefore, combining rapid objective colorimetry with thermal profiling provides a dual-layer verification process, allowing for the simultaneous assessment of both visual consistency and physicochemical formulation stability.
- In thermogravimetric experiment, how many tests for each Viagra tablet sample? n should not be at least below 3 for all analytical experiments;
Thank you for this important question regarding the statistical validity of our data. We confirm that all thermal analysis experiments (TG, DTG, c-DTA) were performed in triplicate (n=3) for each sample to verify the reproducibility of the thermal degradation profiles. The curves presented in the figures are representative of these repeated measurements.
We apologize for not explicitly stating the number of replicates for the thermal methods in the original manuscript. We have now updated Section 2.2.3 (Analysis procedure) to include this information clearly.
Added text:
"Each measurement was repeated three times (n=3) under identical conditions to ensure the reproducibility of the results. The TG, DTG, and c-DTA curves presented in this study are representative of the obtained datasets."
- The quality of counterfeit Viagra® tablets should also verify standard methods
Thank you for this insightful comment. We fully acknowledge that standard pharmacopoeial methods, such as HPLC or dissolution testing, are the gold standard for quantitative quality control and precise determination of the active ingredient (API) content.
However, the primary aim of this study was to evaluate the utility of thermal (TG, DTG, c-DTA) and colorimetric methods as rapid, solvent-free screening tools specifically for distinguishing authentic products from counterfeits based on their physical properties and excipient composition, rather than solely on API quantification.
We deliberately focused on these methods because:
- Detection of excipient differences: counterfeit tablets often contain the correct amount of sildenafil (to pass simple chemical tests) but use non-standard, potentially unsafe excipients. Thermal analysis is superior in detecting these "matrix" differences, which standard HPLC analysis focused on the API might overlook.
- Speed and simplicity: the proposed methods allow for measurements without complex sample preparation (e.g., dissolution), which is crucial for rapid field screening by customs or law enforcement.
Therefore, we treated the standard methods as a separate analytical tier. To address your valid point, we have added a statement in the Study Limitations section acknowledging that while our methods effectively detect formulation anomalies, they should ideally be complemented by standard quantitative techniques for a full toxicological profile.
We have updated the "Study limitations" section with the following sentence:
It should be noted that while thermal and colorimetric analyses effectively detect formulation inconsistencies and differences in excipient composition, they serve primarily as rapid screening tools. For a complete quality verification, particularly regarding the precise quantification of the active substance dose, these methods should ideally be complemented by standard pharmacopoeial techniques such as HPLC.
- It should establish the standard parameters for discriminating original Viagra® tablets and counterfeit Viagra® tablets based the combined method.
Thank you for this suggestion. We agree that summarizing the specific cut-off parameters for authenticity is crucial for the practical application of our findings.
Based on the data obtained for the original reference product (V0), we have defined a "Fingerprint Profile" that serves as the standard for discrimination. In the revised manuscript (Conclusion section), we have explicitly listed these parameters. Any sample deviating from these ranges is classified as non-authentic.
The established standard parameters for the combined method are:
- Colorimetric Criterion: A total color difference E* < 1.5 relative to the reference standard (indicating a barely visible difference).
- Thermal Stability (TG): A total mass loss strictly within the range of 48–50% (Reference V0 = 48.86%).
- Decomposition Profile (DTG): The mandatory presence of a fourth decomposition stage at approx. 356°C, characteristic of the specific excipient matrix (microcrystalline cellulose/magnesium stearate) used in the genuine formulation.
We have added a paragraph to the Conclusions section to formalize these criteria.
Based on the obtained results, we propose the following standard parameters for discriminating authentic Viagra® tablets using the combined method: (1) a colorimetric difference of E* < 1.5 relative to the reference; (2) a total mass loss in TG analysis within the narrow range of 48–50%; and (3) the presence of a specific DTG decomposition peak at approximately 356oC. Samples failing to meet these simultaneous criteria (e.g., V2–V8) should be classified as potential counterfeits.
Round 2
Reviewer 1 Report
Comments and Suggestions for Authors
The authors did all required recommendations. I appreciate their responses. The paper could be published in the current form.
Reviewer 2 Report
Comments and Suggestions for Authors
The authors have addressed most of my concerns.